# Escaping from the Barren Plateau via Gaussian Initializations in Deep Variational Quantum Circuits

Kaining Zhang*, Liu Liu*, Min-Hsiu Hsieh†, and Dacheng Tao*

*School of Computer Science, Faculty of Engineering, The University of Sydney, Sydney, Australia
†Hon Hai Quantum Computing Research Center, Taipei, Taiwan

## Abstract

Variational quantum circuits have been widely employed in quantum simulation and quantum machine learning in recent years. However, quantum circuits with random structures have poor trainability due to the exponentially vanishing gradient with respect to the circuit depth and the qubit number. This result leads to a general standpoint that deep quantum circuits would not be feasible for practical tasks. In this work, we propose an initialization strategy with theoretical guarantees for the vanishing gradient problem in general deep quantum circuits. Specifically, we prove that under proper Gaussian initialized parameters, the norm of the gradient decays at most polynomially when the qubit number and the circuit depth increase. Our theoretical results hold for both the local and the global observable cases, where the latter was believed to have vanishing gradients even for very shallow circuits. Experimental results verify our theoretical findings in quantum simulation and quantum chemistry.

## 1  Introduction

Quantum computing has attracted great attention in recent years, especially since the realization of quantum supremacy [1, 2] with noisy intermediate-scale quantum (NISQ) devices [3]. Due to mild requirements on the gate noise and the circuit connectivity, variational quantum algorithms (VQAs) [4] become one of the most promising frameworks for achieving practical quantum advantages on NISQ devices. Specifically, different VQAs have been proposed for many topics, e.g., quantum chemistry [5–13], quantum simulations [14–23], machine learning [24–31], numerical analysis [32–36], and linear algebra problems [37–39]. Recently, various small-scale VQAs have been implemented on real quantum computers for tasks such as finding the ground state of molecules [10–12] and exploring applications in supervised learning [25], generative learning [30] and reinforcement learning [29].

Typical variational quantum algorithms is a trainable quantum-classical hybrid framework based on parameterized quantum circuits (PQCs) [40]. Similar to classical counterparts such as neural networks [41], first-order methods including the gradient descent [42] and its variants [43] are widely employed in optimizing the loss function of VQAs. However, VQAs may face the trainability barrier when scaling up the size of quantum circuits (i.e., the number of involved qubits or the circuit depth), which is known as the barren plateau problem [44].

Roughly speaking, the barren plateau describes the phenomenon that the value of the loss function and its gradients concentrate around their expectation values with exponentially small variances. We remark that gradient-based methods could hardly handle trainings with the barren plateau phenomenon [45]. Both the machine noise of the quantum channel and the statistical noise induced by measurements could severely degrade the estimation of gradients. Moreover, the optimization of the loss with a flat surface takes much more time using inaccurate gradients than ideal cases. Thus, solving the barren plateau problem is imperative for achieving practical quantum advantages with

VQAs. In this paper, we propose Gaussian initializations for VQAs which have theoretical guarantees on the trainability. We prove that for Gaussian initialized parameters with certain variances, the expectation of the gradient norm is lower bounded by the inverse of the polynomial term of the qubit number and the circuit depth. Technically, we consider various cases regarding VQAs in practice, which include local or global observables, independently or jointly employed parameters, and noisy optimizations induced by finite measurements. To summarize, our contributions are fourfold:

- We propose a Gaussian initialization strategy for deep variational quantum circuits. By setting the variance $\gamma^2 = \mathcal{O}(\frac{1}{L})$ for $N$-qubit $L$-depth circuits with independent parameters and local observables, we lower bound the expectation of the gradient norm by $\mathrm{poly}(N, L)^{-1}$ as provided in Theorem 4.1, which outperforms previous $2^{-\mathcal{O}(L)}$ results.

- We extend the gradient norm result to the global observable case in Theorem 4.2, which was believed to have the barren plateau problem even for very shallow circuits. Moreover, our bound holds for correlated parameterized gates, which are widely employed in practical tasks like quantum chemistry and quantum simulations.

- We provide further analysis on the number of necessary measurements for estimating the gradient, where the noisy case differs from the ideal case with a Gaussian noise. The result is presented in Corollary 4.3, which proves that $\mathcal{O}(\frac{L}{\epsilon})$ times of measurement is sufficient to guarantee a large gradient.

- We conduct various numerical experiments including finding the ground energy and the ground state of the Heisenberg model and the LiH molecule, which belong to quantum simulation and quantum chemistry, respectively. Experiment results show that Gaussian initializations outperform uniform initializations, which verify proposed theorems.

## 1.1 Related work

The barren plateau phenomenon was first noticed in [44], which proves that if the circuit distribution forms unitary 2-designs [46], the variance of the gradient of the circuit vanishes to zero with the rate exponential in the qubit number. Subsequently, several positive results are proved for shallow quantum circuits such as the alternating-layered circuit [45, 47] and the quantum convolutional neural network [48] when the observable is constrained in small number of qubits (local observable). For shallow circuits with $N$ qubits and $\mathcal{O}(\log N)$ depth, the variance of the gradient has the order $\mathrm{poly}(N)^{-1}$ if gate blocks in the circuit are sampled from local 2-design distributions. Later, several works prove an inherent relationship between the barren plateau phenomenon and the complexity of states generated from the circuit. Specifically, circuit states that satisfy the volume law could lead to the barren plateau problem [49]. Expressive quantum circuits, which is measured by the distance between the Haar distribution and the distribution of circuit states, could have vanishing gradients [50]. Since random circuits form approximately 2-designs when they achieve linear depths [46], deep quantum circuits were believed to suffer the barren plateau problem generally.

The parameterization of quantum circuits is achieved by tuning the time of Hamiltonian simulations, so the gradient of the circuit satisfies the parameter-shift rule [51]. Thus, the variance of the loss in VQAs and that of its gradient have similar behaviors for uniform distributions [44, 52]. One corollary of the parameter-shift rule is that the gradient of depolarized noisy quantum circuits vanishes exponentially with increasing circuit depth [53], since the loss itself vanishes in the same rate. Another corollary is that both gradient-free [54] and higher-order methods [55] could not solve the barren plateau problem. Although most existing theoretical and practical results imply the barren plateau phenomenon in deep circuits, VQAs with deep circuits do have impressive advantages from other aspects. For example, the loss of VQAs is highly non-convex, which is hard to find the global minima [56] for both shallow and deep circuits. Meanwhile, for VQAs with shallow circuits, local minima and global minima have considerable gaps [57], which could severely influence the training performance of gradient-based methods. Contrary to shallow cases, deep VQAs have vanishing gaps between local minima and global minima [58]. In practice, experiments show that overparameterized VQAs [59] can be optimized towards the global minima. Moreover, VQAs with deep circuits have more expressive power than that of shallow circuits [60–62], which implies the potential to handle more complex tasks in quantum machine learning and related fields.

Inspired by various advantages of deep VQAs, some approaches have been proposed recently for solving the related barren plateau problem in practice [63–65]. For example, the block-identity

strategy [63] initializes gate blocks in pairs and sets parameters inversely, such that the initial circuit is equivalent to the identity circuit with zero depth. Since shallow circuits have no vanishing gradient problem, the corresponding VQA is trainable with guarantees at the first step. However, we remark that the block-identity condition would not hold after the first step, and the structure of the circuit needs to be designed properly. The layerwise training method [64] trains parameters in the circuit layers by layers, such that the depth of trainable part is limited. However, this method implements circuits with larger depth than that of the origin circuit, and parameters in the first few layers are not optimized. A recent work provides theoretical guarantees on the trainability of deep circuits with certain structures [65]. However, the proposed theory only suits VQAs with local observables, but many practical applications such as finding the ground state of molecules and the quantum compiling [66, 67] apply global observables.

## 2 Notations and quantum computing basics

We denote by $[N]$ the set $\{1, \cdots, N\}$. The form $\| \cdot \|_2$ represents the $\ell_2$ norm for the vector and the spectral norm for the matrix, respectively. We denote by $a_j$ the $j$-th component of the vector $\boldsymbol{a}$. The tensor product operation is denoted as "$\otimes$". The conjugate transpose of a matrix $A$ is denoted as $A^\dagger$. The trace of a matrix $A$ is denoted as $\text{Tr}[A]$. We denote $\nabla_{\boldsymbol{\theta}} f$ as the gradient of the function $f$ with respect to the variable $\boldsymbol{\theta}$. We employ notations $\mathcal{O}$ to describe complexity notions.

Now we introduce quantum computing knowledge and notations. The pure state of a qubit could be written as $|\phi\rangle = a|0\rangle + b|1\rangle$, where $a, b \in \mathbb{C}$ satisfy $|a|^2 + |b|^2 = 1$, and $|0\rangle = (1, 0)^T, |1\rangle = (0, 1)^T$. The $N$-qubit space is formed by the tensor product of $N$ single-qubit spaces. For pure states, the corresponding density matrix is defined as $\rho = |\phi\rangle\langle\phi|$, in which $\langle\phi| = (|\phi\rangle)^\dagger$. We use the density matrix to represent general mixed quantum states, i.e., $\rho = \sum_k c_k |\phi_k\rangle\langle\phi_k|$, where $c_k \in \mathbb{R}$ and $\sum_k c_k = 1$. A single-qubit operation to the state behaves like the matrix-vector multiplication and can be referred to as the gate —□— in the quantum circuit language. Specifically, single-qubit operations are often used as $R_X(\theta) = e^{-i\theta X}$, $R_Y(\theta) = e^{-i\theta Y}$, and $R_Z(\theta) = e^{-i\theta Z}$, where

$$X = \begin{pmatrix} 0 & 1 \\ 1 & 0 \end{pmatrix}, Y = \begin{pmatrix} 0 & -i \\ i & 0 \end{pmatrix}, Z = \begin{pmatrix} 1 & 0 \\ 0 & -1 \end{pmatrix}.$$

Pauli matrices will be referred to as $\{I, X, Y, Z\} = \{\sigma_0, \sigma_1, \sigma_2, \sigma_3\}$ for the convenience. Moreover, two-qubit operations, such as the CZ gate and the $\sqrt{i\text{SWAP}}$ gate, are employed for generating quantum entanglement:

$$\text{CZ} = \begin{pmatrix} 1 & 0 & 0 & 0 \\ 0 & 1 & 0 & 0 \\ 0 & 0 & 1 & 0 \\ 0 & 0 & 0 & -1 \end{pmatrix}, \sqrt{i\text{SWAP}} = \begin{pmatrix} 1 & 0 & 0 & 0 \\ 0 & 1/\sqrt{2} & i/\sqrt{2} & 0 \\ 0 & i/\sqrt{2} & 1/\sqrt{2} & 0 \\ 0 & 0 & 0 & 1 \end{pmatrix}.$$

We could obtain information from the quantum system by performing measurements, for example, measuring the state $|\phi\rangle = a|0\rangle + b|1\rangle$ generates 0 and 1 with probability $p(0) = |a|^2$ and $p(1) = |b|^2$, respectively. Such a measurement operation could be mathematically referred to as calculating the average of the observable $O = \sigma_3$ under the state $|\phi\rangle$:

$$\langle\phi|O|\phi\rangle \equiv \text{Tr}[\sigma_3 |\phi\rangle\langle\phi|] = |a|^2 - |b|^2 = p(0) - p(1).$$

Mathematically, quantum observables are Hermitian matrices. Specifically, the average of a unitary observable under arbitrary states is bounded by $[-1, 1]$. We remark that $\mathcal{O}(\frac{1}{\epsilon^2})$ times of measurements could provide an $\epsilon\|O\|_2$-error estimation to the value $\text{Tr}[O\rho]$.

## 3 Framework of general VQAs

In this section, we introduce the framework of general VQAs and corresponding notations. A typical variational quantum algorithm can be viewed as the optimization of the function $f$, which is defined as the expectation of observables. The expectation varies for different initial states and different parameters $\boldsymbol{\theta}$ used in quantum circuits. Throughout this paper, we define

$$f(\boldsymbol{\theta}) = \text{Tr}\left[OV(\boldsymbol{\theta})\rho_{\text{in}}V(\boldsymbol{\theta})^\dagger\right] \tag{1}$$

as the loss function of VQAs, where $V(\boldsymbol{\theta})$ denotes the parameterized quantum circuit, the hermitian matrix $O$ denotes the observable, and $\rho_{\text{in}}$ denotes the density matrix of the input state. Next, we explain observables, input states, and parameterized quantum circuits in detail.

Both the observable and the density matrix could be decomposed under the Pauli basis. We define the *locality* of a quantum observable as the maximum number of non-identity Pauli matrices in the tensor product, such that the corresponding coefficient is not zero. Thus, the observable with the constant locality is said to be *local*, and the observable that acts on all qubits is said to be *global*.

The observable and the input state in VQAs could have various formulations for specific tasks. For the quantum simulation or the quantum chemistry scenario, observables are constrained to be the system Hamiltonians, while input states are usually prepared as computational basis states. For example, $(|0\rangle\langle 0|)^{\otimes N}$ is used frequently in quantum simulations [17, 18]. Hartree–Fock (HF) states [9, 10], which are prepared by the tensor product of $\{|0\rangle, |1\rangle\}$, serve as good initial states in quantum chemistry tasks [9, 11–13]. For quantum machine learning (QML) tasks, initial states encode the information of the training data, which could have a complex form. Many encoding strategies have been introduced in the literature [24, 68, 69]. In contrary with the complex initial states, observables employed in QML are quite simple. For example, $\pi_0 = |0\rangle\langle 0|$ serves as the observable in most QML tasks related with the classification [24–26] or the dimensional reduction [70].

Apart from the input states and the observable choices, parameterized quantum circuits employed in different variational quantum algorithms have various structures, which are also known as *ansatzes* [71–73]. Specifically, the ansatz in the VQA denotes the initial guess on the circuit structure. For example, alternating-layered ansatzes [71, 74] are proposed for approximating the Hamiltonian evolution. Recently, hardware efficient ansatzes [7, 75] and tensor-network based ansatzes [76, 77], which could utilize parameters efficiently on noisy quantum computers, have been developed for various tasks, including quantum simulations and quantum machine learning. For quantum chemistry tasks, unitary coupled cluster ansatzes [78, 79] are preferred since they preserve the number of electrons corresponding to circuit states.

In practice, ansatz is deployed as the sequence of single-qubit rotations $\{e^{-i\theta\sigma_k}, k \in \{1, 2, 3\}\}$ and two-qubit gates. We remark that the gradient of the VQA satisfies the parameter-shift rule [51, 80, 81]; namely, for independently deployed parameters $\theta_j$, the corresponding partial derivative is

$$\frac{\partial f}{\partial \theta_j} = f(\boldsymbol{\theta}_+) - f(\boldsymbol{\theta}_-), \tag{2}$$

where $\boldsymbol{\theta}_+$ and $\boldsymbol{\theta}_-$ are different from $\boldsymbol{\theta}$ only at the $j$-th parameter: $\theta_j \to \theta_j \pm \frac{\pi}{4}$. Thus, the gradient of $f$ could be estimated efficiently, which allows optimizing VQAs with gradient-based methods [82–84].

## 4 Theoretical results about Gaussian initialized VQAs

In this section, we provide our theoretical guarantees on the trainability of deep quantum circuits through proper designs for the initial parameter distribution. In short, we prove that the gradient of the $L$-layer $N$-qubit circuit is upper bounded by $1/\text{poly}(L, N)$, if initial parameters are sampled from a Gaussian distribution with $\mathcal{O}(1/L)$ variance. Our bounds significantly improve existing results of the gradients of VQAs, which have the order $2^{-\mathcal{O}(L)}$ for shallow circuits and the order $2^{-\mathcal{O}(N)}$ for deep circuits. We prove different results for the local and global observable cases in Section 4.1 and Section 4.2, respectively.

### 4.1 Independent parameters with local observables

First, we introduce the Gaussian initialization of parameters for the local observable case. We use the quantum circuit illustrated in Figure 1 as the ansatz in this section. The circuit in Figure 1 performs $L$ layers of single qubit rotations and CZ gates on the input state $\rho_{\text{in}}$, followed by a $R_X$ layer and a $R_Y$ layer. We denote the single-qubit gate on the $n$-th qubit of the $\ell$-th layer as $e^{-i\theta_{\ell,n} G_{\ell,n}}$, $\forall \ell \in \{1, \cdots, L+2\}$ and $n \in \{1, \cdots, N\}$, where $\theta_{\ell,n}$ is the corresponding parameter and $G_{\ell,n}$ is a Hermitian unitary. To eliminate degenerate parameters, we require that single-qubit gates in the first $L$ layers do not commute with the CZ gate. After gates operations, we measure the observable

$$\sigma_{\boldsymbol{i}} = \sigma_{(i_1, i_2, \cdots, i_N)} = \sigma_{i_1} \otimes \sigma_{i_2} \otimes \cdots \otimes \sigma_{i_N}, \tag{3}$$

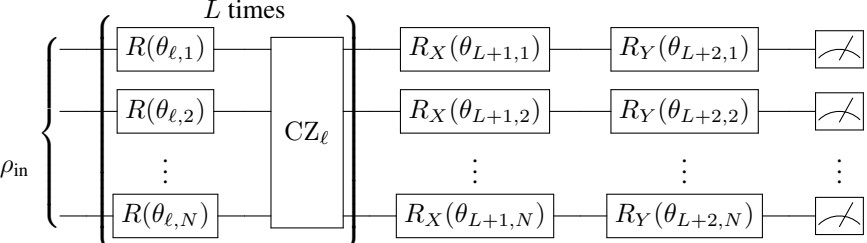

Figure 1: The quantum circuit framework for the local observable case. The circuit performs $L$ layers of single qubit rotations and CZ layers on the input state $\rho_{\text{in}}$, followed by a $R_X$ layer and a $R_Y$ layer. In the $\ell$-th single qubit layer, we employ the gate $e^{-i\theta_{\ell,n}G_{\ell,n}}$ for all qubits $n \in [N]$, where $G_{\ell,n}$ is a Hermitian unitary, which anti-commutes with $\sigma_3$ for $\ell \in [L]$. In each $\text{CZ}_\ell$ layer, CZ gates are employed between arbitrary qubit pairs. The measurement is performed on $S$ qubits where the observable acts nontrivially on these qubits.

where $i_j \in \{0, 1, 2, 3\}, \forall j \in \{1, \cdots, N\}$, and $\boldsymbol{i}$ contains $S$ non-zero elements. Figure 1 provides a general framework of VQAs with local observables, which covers various ansatzes proposed in the literature [65, 85, 61, 64]. The bound of the gradient norm of the Gaussian initialized variational quantum circuit is provided in Theorem 4.1 with the proof in Appendix.

**Theorem 4.1.** *Consider the $L$-layer $N$-qubit variational quantum circuit $V(\boldsymbol{\theta})$ defined in Figure 1 and the cost function $f(\boldsymbol{\theta}) = \text{Tr}\left[\sigma_{\boldsymbol{i}} V(\boldsymbol{\theta}) \rho_{\text{in}} V(\boldsymbol{\theta})^\dagger\right]$, where the observable $\sigma_{\boldsymbol{i}}$ follows the definition (3). Then,*

$$\mathbb{E}_{\boldsymbol{\theta}} \|\nabla_{\boldsymbol{\theta}} f\|^2 \geq \frac{L}{S^S(L+2)^{S+1}} \text{Tr}\left[\sigma_{\boldsymbol{j}} \rho_{\text{in}}\right]^2, \tag{4}$$

*where $S$ is the number of non-zero elements in $\boldsymbol{i}$, and the index $\boldsymbol{j} = (j_1, j_2, \cdots, j_N)$ such that $j_m = 0, \forall i_m = 0$ and $j_m = 3, \forall i_m \neq 0$. The expectation is taken with the Gaussian distribution $\mathcal{N}\left(0, \frac{1}{4S(L+2)}\right)$ for the parameters $\boldsymbol{\theta}$.*

Compared to existing works [44, 45, 47, 48, 65], Theorem 4.1 provides a larger lower bound of the gradient norm, which improves the complexity exponentially with the depth of trainable circuits. Different from unitary 2-design distributions [44, 45, 47, 48] or the uniform distribution in the parameter space [52, 86, 65] that were employed in existing works, we analyze the expectation of the gradient norm under a depth-induced Gaussian distribution. This change follows a natural idea that the trainability is not required in the whole parameter space or the entire circuit space, but only on the parameter trajectory during the training. Moreover, large norm of gradients could only guarantee the trainability in the beginning stage, instead of the whole optimization, since a large gradient for trained parameters corresponds to non-convergence. Thus, the barren plateau problem could be crucial if initial parameters have vanishing gradients, which has been proved for deep VQAs with uniform initializations. In contrary, we could solve the barren plateau problem if parameters are initialized properly with large gradients, as provided in Theorem 4.1. Finally, Gaussian initialized circuits converge to benign values if optima appear around $\boldsymbol{\theta} = \boldsymbol{0}$, which holds in many cases. For example, over-parameterized quantum circuits have benign local minima [58] if the number of parameters exceeds the over-parameterization threshold. Moreover, over-parameterized circuits have exponential convergence rates [87, 88] on tasks like quantum machine learning and the quantum eigensolver. These works indicate that quantum circuits with sufficient depths could find good optimums near the initial points, which is similar to the classical wide neural network case [89].

## 4.2 Correlated parameters with global observables

Next, we extend the Gaussian initialization framework to general quantum circuits with correlated parameters and global observables. Quantum circuits with correlated parameters have wide applications

in quantum simulations and quantum chemistry [9, 11–13]. One example is the Givens rotation

$$
R^{\text{Givens}}(\theta) = \begin{pmatrix} 1 & 0 & 0 & 0 \\ 0 & \cos\theta & -\sin\theta & 0 \\ 0 & \sin\theta & \cos\theta & 0 \\ 0 & 0 & 0 & 1 \end{pmatrix} = \quad \tag{5}
$$

which preserves the number of electrons in parameterized quantum states [11].

To analyze VQAs with correlated parameterized gates, we consider the ansatz $V(\boldsymbol{\theta}) = \prod_{j=L}^{1} V_j(\theta_j)$, which consists of parameterized gates $\{V_j(\theta_j)\}_{j=1}^{L}$. Denote by $h_j$ the number of unitary gates that share the same parameter $\theta_j$. Thus, the parameterized gate $V_j(\theta_j)$ consists of a list of fixed and parameterized unitary operations

$$
V_j(\theta_j) = \prod_{k=1}^{h_j} W_{jk} e^{-i \frac{\theta_j}{a_j} G_{jk}} \tag{6}
$$

with the term $a_j \in \mathbb{R}/\{0\}$, where the Hamiltonian $G_{jk}$ and the fixed gate $W_{jk}$ are unitary $\forall k \in [h_j]$. Moreover, we consider the objective function

$$
f(\boldsymbol{\theta}) = \text{Tr}\left[ O \prod_{j=L}^{1} V_j(\theta_j) \rho_{\text{in}} \prod_{j=1}^{L} V_j(\theta_j)^\dagger \right], \tag{7}
$$

where $\rho_{\text{in}}$ and $O$ denote the input state and the observable, respectively. In practical tasks of quantum chemistry, the molecule Hamiltonian $H$ serves as the observable $O$. Minimizing the function (7) provides the ground energy and the corresponding ground state of the molecule. We provide the bound of the gradient norm of the Gaussian initialized variational quantum circuit in Theorem 4.2 with the proof in Appendix. Similar to the local observable case, we could bound the norm of the gradient of Eq. (7) if parameters are initialized with $\mathcal{O}(\frac{1}{L})$ variance. Theorem 4.2 provides nontrivial bounds when the gradient at the zero point is large. This condition holds when the mean-field theory provides a good initial guess to the corresponding problems, e.g. the ground energy task in quantum chemistry and quantum many-body problems [90].

**Theorem 4.2.** *Consider the $N$-qubit variational quantum algorithms with the objective function (7). Then the following formula holds for any $\ell \in \{1, \cdots, L\}$,*

$$
\mathbb{E}_{\boldsymbol{\theta}} \left( \frac{\partial f}{\partial \theta_\ell} \right)^2 \geq (1 - \epsilon) \left( \frac{\partial f}{\partial \theta_\ell} \right)^2 \Bigg|_{\boldsymbol{\theta} = \boldsymbol{0}}, \tag{8}
$$

*where $\boldsymbol{0} \in \mathbb{R}^L$ is the zero vector. The expectation is taken with Gaussian distributions $\mathcal{N}(0, \gamma_j^2)$ for parameters in $\boldsymbol{\theta} = \{\theta_j\}_{j=1}^{L}$, where the variance $\gamma_j^2 \leq \frac{a_j^2 \epsilon}{16 h_j^2 (3 h_j (h_j - 1) + 1) L \|O\|_2^2} \left( \frac{\partial f}{\partial \theta_\ell} \right)^2 \Bigg|_{\boldsymbol{\theta} = \boldsymbol{0}}.$*

We remark that Theorem 4.2 not only provides an initialization strategy, but also guarantees the update direction during the training. Different from the classical neural network, where the gradient could be calculated accurately, the gradient of VQAs, obtained by the parameter-shift rule (2), is perturbed by the measurement noise. A guide on the size of acceptable measurement noise could be useful for the complexity analysis of VQAs. Specifically, define $\boldsymbol{\theta}^{(t-1)}$ as the parameter at the $t - 1$-th iteration. Denote by $\boldsymbol{\theta}^{(t)}$ and $\tilde{\boldsymbol{\theta}}^{(t)}$ the parameter updated from $\boldsymbol{\theta}^{(t-1)}$ for noiseless and noisy cases, respectively. Then $\tilde{\boldsymbol{\theta}}^{(t)}$ differs from $\boldsymbol{\theta}^{(t)}$ by a Gaussian error term due to the measurement noise. We expect to derive the gradient norm bound for $\tilde{\boldsymbol{\theta}}^{(t)}$, as provided in Corollary 4.3. Thus, $\frac{1}{\gamma^2} = \mathcal{O}(\frac{L}{\epsilon})$ number of measurements is sufficient to guarantee a large gradient.

**Corollary 4.3.** *Consider the $N$-qubit variational quantum algorithms with the objective function (7). Then the following formula holds for any $\ell \in \{1, \cdots, L\}$,*

$$
\mathbb{E}_{\boldsymbol{\delta}} \left( \frac{\partial f}{\partial \theta_\ell} \right)^2 \Bigg|_{\boldsymbol{\theta} = \boldsymbol{\theta}^{(t)} + \boldsymbol{\delta}} \geq (1 - \epsilon) \left( \frac{\partial f}{\partial \theta_\ell} \right)^2 \Bigg|_{\boldsymbol{\theta} = \boldsymbol{\theta}^{(t)}}. \tag{9}
$$

*The expectation is taken with Gaussian distributions $\mathcal{N}(0, \gamma_j^2)$ for parameters $\boldsymbol{\delta} = \{\delta_j\}_{j=1}^{L}$, where the variance $\gamma_j^2 \leq \frac{a_j^2 \epsilon}{16 h_j^2 (3 h_j (h_j - 1) + 1) L \|O\|_2^2} \left( \frac{\partial f}{\partial \theta_\ell} \right)^2 \Bigg|_{\boldsymbol{\theta} = \boldsymbol{\theta}^{(t)}}, \forall j \in [L].$*

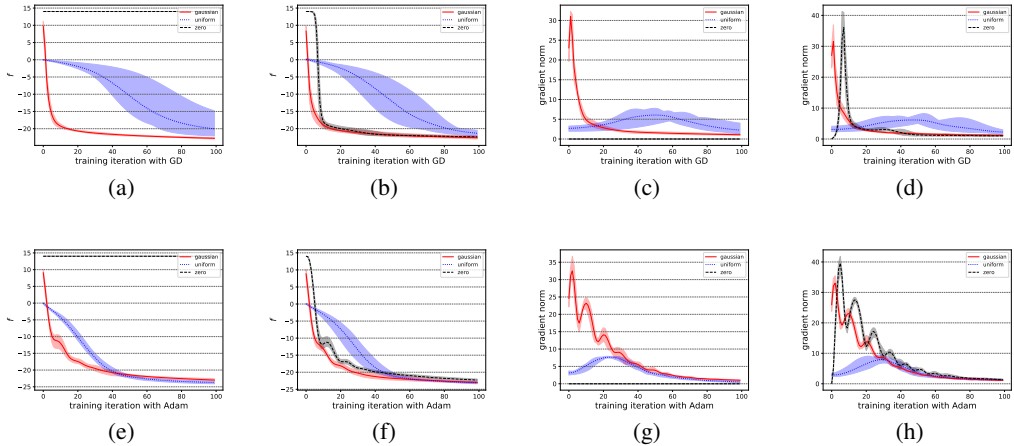

Figure 2: Numerical results of finding the ground energy of the Heisenberg model. The first row shows training results with the gradient descent optimizer, where Figures 2(a) and 2(b) illustrate the loss function corresponding to Eq.(10) during the optimization with accurate and noisy gradients, respectively. Figures 2(c) and 2(d) show the $\ell_2$ norm of corresponding gradients. The second row shows training results with the Adam optimizer, where Figures 2(e) and 2(f) illustrate the loss function with accurate and noisy gradients, respectively. Figures 2(g) and 2(h) show the $\ell_2$ norm of corresponding gradients. Each line denotes the average of 5 rounds of optimizations.

Corollary 4.3 is derived by analyzing the gradient of the function $g(\boldsymbol{\delta}) = f(\boldsymbol{\delta} + \boldsymbol{\theta}^{(t)})$ via Theorem 4.2. For any number of measurements such that the corresponding Gaussian noise $\boldsymbol{\delta}$ satisfies the condition in Corollary 4.3, the trainability at the updated point is guaranteed.

## 5 Experiments

In this section, we analyze the training behavior of two variational quantum algorithms, i.e., finding the ground energy and state of the Heisenberg model and the LiH molecule, respectively. All numerical experiments are provided using the Pennylane package [91].

### 5.1 Heisenberg model

In the first task, we aim to find the ground state and the ground energy of the Heisenberg model [92]. The corresponding Hamiltonian matrix is

$$H = \sum_{i=1}^{N-1} X_i X_{i+1} + Y_i Y_{i+1} + Z_i Z_{i+1}, \tag{10}$$

where $N$ is the number of qubit, $X_i = I^{\otimes(i-1)} \otimes X \otimes I^{\otimes(N-i)}$, $Y_i = I^{\otimes(i-1)} \otimes Y \otimes I^{\otimes(N-i)}$, and $Z_i = I^{\otimes(i-1)} \otimes Z \otimes I^{\otimes(N-i)}$. We employ the loss function defined by Eq. (1) with the input state $(|0\rangle\langle 0|)^{\otimes N}$ and the observable (10). Thus, by minimizing the function (1), we can obtain the least eigenvalue of the observable (10), which is the ground energy. We adopt the ansatz with $N = 15$ qubits, which consists of $L_1 = 10$ layers of $R_Y R_X CZ$ blocks. In each block, we first employ the CZ gate to neighboring qubits pairs $\{(1,2)\cdots,(N,1)\}$, followed by $R_X$ and $R_Y$ rotations for all qubits. Overall, the quantum circuit has 300 parameters. We consider three initialization methods for comparison, i.e., initializations with the Gaussian distribution $\mathcal{N}(0, \gamma^2)$ and the uniform distribution in $[0, 2\pi]$, respectively, and the zero initialization (all parameters equal to 0 at the initial point). We remark that each term in the observable (10) contains at most $S = 2$ non-identity Pauli matrices, which is consistent with the $(S, L) = (2, 18)$ case of Theorem 4.1. Thus, we expect that the Gaussian initialization with the variance $\gamma^2 = \frac{1}{4S(L+2)} = \frac{1}{160}$ could provide trainable initial parameters.

In the experiment, we train VQAs with gradient descent (GD) [93] and Adam optimizers [94], respectively. The learning rate is 0.01 and 0.01 for both GD and Adam cases. Since the estimation

of gradients on real quantum computers could be perturbed by statistical measurement noise, we compare optimizations using accurate and noisy gradients. For the latter case, we set the variance of measurement noises to be $0.01$. The numerical results of the Heisenberg model are shown in the Figure 2. The loss during the training with gradient descents is shown in Figures 2(a) and 2(b) for the accurate and the noisy gradient cases, respectively. The Gaussian initialization outperforms the other two initializations with faster convergence rates. Figures 2(c) and 2(d) verify that Gaussian initialized VQAs have larger gradients in the early stage, compared to that of uniformly initialized VQAs. We notice that zero initialized VQAs cannot be trained with accurate gradient descent, since the initial gradient equals to zero. This problem is alleviated in the noisy case, as shown in Figures 2(b) and 2(d). Since the gradient is close to zero at the initial stage, the update direction mainly depends on the measurement noise, which forms the Gaussian distribution. Thus, the parameter in the noisy zero initialized VQAs is expected to accumulate enough variances, which takes around 10 iterations based on Figure 2(h). As illustrated in Figure 2(b), the loss function corresponding to the zero initialization decreases quickly after the variance accumulation stage. Results in Figures 2(e) and 2(h) show similar training behaviors using the Adam optimizer.

## 5.2 Quantum chemistry

In the second task, we aim to find the ground state and the ground energy of the LiH molecule. We follow settings on the ansatz in Refs. [12, 13]. For the molecule with $n_e$ active electrons and $n_o$ free spin orbitals, the corresponding VQA contains $N = n_o$ qubits, which employs the HF state [9, 10]

$$|\phi_{\mathrm{HF}}\rangle = \underbrace{|1\rangle \otimes \cdots |1\rangle}_{n_e} \otimes \underbrace{|0\rangle \otimes \cdots |0\rangle}_{n_o - n_e}$$

as the input state. We construct the parameterized quantum circuit with Givens rotation gates [12], where each gate is implemented on 2 or 4 qubits with one parameter. Specifically, for the LiH molecule, the number of electrons $n_e = 2$, the number of free spin orbitals $n_o = 10$, and the number of different Givens rotations is $L = 24$ [13]. We follow the molecule Hamiltonian $H_{\mathrm{LiH}}$ defined in Ref. [13]. Thus, the loss function for finding the ground energy of LiH is defined as

$$f(\boldsymbol{\theta}) = \mathrm{Tr}\left[ H_{\mathrm{LiH}} V_{\mathrm{Givens}}(\boldsymbol{\theta}) |\phi_{\mathrm{HF}}\rangle\langle\phi_{\mathrm{HF}}| V_{\mathrm{Givens}}(\boldsymbol{\theta})^\dagger \right], \tag{11}$$

where $V_{\mathrm{Givens}}(\boldsymbol{\theta}) = \prod_{i=1}^{24} R_i^{\mathrm{Givens}}(\theta_i)$ denotes the product of all parameterized Givens rotations of the LiH molecule. By minimizing the function (11), we can obtain the least eigenvalue of the Hamiltonian $H_{\mathrm{LiH}}$, which is the ground energy of the LiH molecule.

In practice, we initialize parameters in the VQA (11) with three distributions for comparison, i.e., the Gaussian distribution $\mathcal{N}(0, \gamma^2)$, the zero distribution (all parameters equal to 0), and the uniform distribution in $[0, 2\pi]$. For 2-qubit Givens rotations, the term $(h, a) = (2, 2)$ as shown in Eq. (5). For 4-qubit Givens rotations, the term $(h, a) = (8, 8)$ [95]. Thus, we set the variance in the Gaussian distribution $\gamma^2 = \frac{8^2 \times \frac{1}{2}}{48 \times 8^4 \times 24}$, which matches the $(L, h, a, \epsilon) = (24, 8, 8, \frac{1}{2})$ case of Theorem 4.2. Similar to the task of the Heisenberg model, we consider both the accurate and the noisy gradient cases, where the variance of noises in the latter case is the constant $0.001$. Moreover, we consider the noisy case with adaptive noises, where the variance of the noise on each partial derivative $\frac{\partial f}{\partial \theta_\ell}\big|_{\boldsymbol{\theta} = \boldsymbol{\theta}^{(t)}}$ in the $t$-th iteration is

$$\gamma^2 = \frac{1}{96 \times 24 \times 8^2 \|H_{\mathrm{LiH}}\|_2^2} \left( \frac{\partial f}{\partial \theta_\ell} \right)^2 \bigg|_{\boldsymbol{\theta} = \boldsymbol{\theta}^{(t-1)}}. \tag{12}$$

The variance in Eq. (12) matches the $(L, h, a, \epsilon) = (24, 8, 8, \frac{1}{2})$ case of Corollary 4.3 when the VQA is nearly converged:

$$\frac{\partial f}{\partial \theta_\ell}\big|_{\boldsymbol{\theta} = \boldsymbol{\theta}^{(t)}} \approx \frac{\partial f}{\partial \theta_\ell}\big|_{\boldsymbol{\theta} = \boldsymbol{\theta}^{(t-1)}}.$$

In the experiment, we train VQAs with gradient descent and Adam optimizers. Learning rates are set to be $0.1$ and $0.01$ for GD and Adam cases, respectively. The loss (11) during training iterations is shown in Figure 3. Optimization results with gradient descents are shown in Figures 3(a)-3(c) for the accurate gradient case, the adaptive noisy gradient case, and the noisy gradient case with the constant noise variance $0.001$, respectively. The variance of the noise in the adaptive noisy gradient case follows Eq. (12). Figures 3(a) and 3(b) show similar performance, where the loss $f$ with the

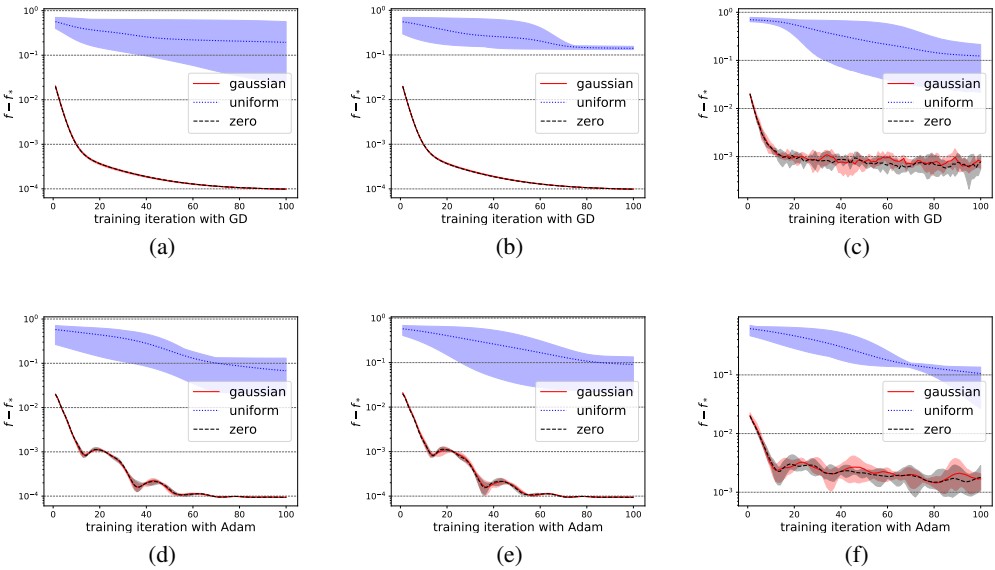

Figure 3: Numerical results of finding the ground energy of the molecule LiH. The first and second rows show training results with the gradient descent and the Adam optimizer, respectively. The left, the middle, and the right columns show results using accurate gradients, noisy gradients with adaptive-distributed noises, and noisy gradients with constant-distributed noises. The variance of noises in the middle line (Figures 3(b) and 3(e)) follows Eq. (12), while the variance of noises in the right line (Figures 3(c) and 3(f)) is 0.001. Each line denotes the average of 5 rounds of optimizations.

Gaussian initialization and the zero initialization converge to $10^{-4}$ over the global minimum $f_*$. The loss with the uniform initialization is higher than $10^{-1}$ over the global minimum. Figure 3(c) shows the training with constantly perturbed gradients. The Gaussian initialization and the zero initialization induce the $10^{-3}$ convergence, while the loss function with the uniform initialization is still higher than $10^{-1}$ over the global minimum. Figures 3(d)-3(f) show similar training behaviors using the Adam optimizer. Based on Figures 3(a)-3(f), the Gaussian initialization and the zero initialization outperform the uniform initialization in all cases. We notice that optimization with accurate gradients and optimization with adaptive noisy gradients have the same convergence rate and the final value of the loss function, which is better than that using constantly perturbed gradients. We remark that the number of measurements $T = \mathcal{O}(\frac{1}{\mathrm{Var(noise)}})$. Thus, finite number of measurements with the noise (12) for gradient estimation is enough to achieve the performance of accurate gradients, which verifies Theorem 4.2 and Corollary 4.3.

## 6 Conclusions

In this work, we provide a Gaussian initialization strategy for solving the vanishing gradient problem of deep variational quantum algorithms. We prove that the gradient norm of $N$-qubit quantum circuits with $L$ layers could be lower bounded by $\mathrm{poly}(N, L)^{-1}$, if the parameter is sampled independently from the Gaussian distribution with the variance $\mathcal{O}(\frac{1}{L})$. Our results hold for both the local and the global observable cases, and could be generalized to VQAs employing correlated parameterized gates. Compared to the local case, the bound for the global case depends on the gradient performance at the zero point. Further analysis towards the zero-case-free bound could be investigated as future directions. Moreover, we show that the necessary number of measurements, which scales $\mathcal{O}(\frac{L}{\epsilon})$, suffices for estimating the gradient during the training. We provide numerical experiments on finding the ground energy and state of the Heisenberg model and the LiH molecule, respectively. Experiments show that the proposed Gaussian initialization method outperforms the uniform initialization method with a faster convergence rate, and the training using gradients with adaptive noises shows the same convergence compared to the training using noiseless gradients.

## Acknowledgement

This work is supported in part by ARC FL-170100117, IC-190100031, and LE-200100049.

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
