# OpenReview forum: "Escaping from the Barren Plateau via Gaussian Initializations in Deep Variational Quantum Circuits"
_NeurIPS.cc/2022/Conference — NeurIPS 2022 Accept_

### Official Review · Reviewer_hWp5 · 2022-06-27

**Rating:** 7
**Confidence:** 4
**Soundness:** 3 good
**Presentation:** 3 good
**Contribution:** 3 good

**Summary:**

The authors demonstrate that over a Gaussian prior of appropriate width, the second moment of the derivatives of a class of variational quantum algorithms (VQAs) is only polynomially small in the problem size $n$ and circuit depth $L$ (for constant problem locality $S$), which is exponentially larger than the traditional barren plateau bounds (taken uniformly over parameters). The authors also give a bound when the locality $S$ grows with the problem size, that lower bounds the expected derivative second moment by (a finite fraction of) the initial squared derivative. The authors then demonstrate numerically that their initialization scheme gives much better optimization performance than uniform initialization in a variety of VQA tasks.

**Questions:**

* I suggest making the previously discussed limitations of the work more clear (in particular, that these results require staying near $\vec{\theta}=\vec{0}$ to hold).
* I suggest providing more intuition as to when Theorem 4.2 and Corollary 4.3 give nontrivial bounds.
* I suggest providing more examples of when the authors' results are expected to hold (e.g. in chemistry where the Hartree--Fock state is a good approximation of the true ground state, etc.).

**Limitations:**

I previously discussed what limitations of the work I believe should be more explicitly mentioned by the authors; namely, I suggest tempering the claims that Gaussian initialization solves all instances of barren plateaus, and providing more examples or intuition as to situations where one might expect the implicit assumptions on training (i.e. staying near $\vec{\theta}=\vec{0}$) to hold.

**Strengths And Weaknesses:**

The introduced bounds are novel, and though I did not check the proofs in complete detail, the authors' technical methods and proofs seem correct. I also enjoyed that these results are essentially a more rigorous understanding of the intuitive fact that training VQAs while near the identity should be similar to low-depth VQAs, and not experience barren plateaus. I think a couple of weaknesses of the paper, though, are maybe claiming too much from the shown results.

First, in their discussion of Theorem 4.2 (and Corollary 4.3), the authors imply that their results are enough to show that Gaussian initialization completely absolves VQAs with global cost functions from barren plateaus. However, these results only lower bound the second moment of the derivative with (a finite fraction of) the initial square of the derivative. This initial derivative can be very small; see, for instance, the global cost function warm-up example in "Cost function dependent barren plateaus in shallow parametrized quantum circuits" (Cerezo et al. 2021), where this initial derivative is zero, giving a trivial bound. I would have enjoyed more discussion (or examples) arguing that this bound is typically only polynomially (not superpolynomially) small.

Second, the authors' results (including now also Theorem 4.1) still rely on assumptions in the training of the model. Namely, once training is far away from the initial $\vec{\theta}=\vec{0}$, it is no longer well-approximated by a Gaussian of polynomially small variance. In fact, for a number of parameters growing superlogarithmically with $n$, roughly the volume of "allowed" region where these results are expected to hold is superpolynomially small in the volume of parameter space (polynomially small in diameter). This (that is, superlogarithmically large depth) is the regime previous barren plateau results kick in (i.e. when averaged uniformly over parameters), and I suspect they may be related. I recommend the authors make this limitation of their work more clear.

These limitations aside, I still find the work a nice, rigorous interpretation of a common approach to circumventing barren plateaus; when one has a good guess for where in parameter space the optimum is (say, near $\vec{\theta}=\vec{0}$), and expect optimization to stay within this region, barren plateaus may be avoided.

---

> ### Author Response · Authors · 2022-08-02
> **Respond to Reviewer hWp5**
>
> We thank the Reviewer for the effort and constructive comments on our paper!
>
> **Q1. I suggest making the previously discussed limitations of the work more clear (in particular, that these results require staying near $\theta=0$ to hold).**
>
> A1: We will add discussions about the limitations of our work in the final version. Specifically, the condition that good optima appear around $\theta=0$ holds in many cases. For example, Ref.[r1] has proved that over-parameterized quantum circuits have benign local minima if the number of parameters exceeds the over-parameterization threshold. Moreover, Refs.[r2,r3] have proved exponential convergence rates for over-parameterized circuits on tasks like quantum machine learning and the quantum eigensolver. These works indicate that quantum circuits with sufficient depths could find good optimums near the initial points, which is similar to the classical wide neural network case [r4]. Furthermore, we remark that the main contribution of this work is a new initialization method for deep quantum circuits that has no barren plateau problem at the initial stages. The current work does not focus on the training dynamic or the local minima property of quantum circuits. However, both concerns lead to meaningful future directions.
>
> **Q2. I suggest providing more intuition as to when Theorem 4.2 and Corollary 4.3 give nontrivial bounds.**
>
> A2: We will refine the current discussion clearly in the final version. In principle, Theorem 4.2 could be helpful if the gradient at the zero point is large. Besides, Theorem 4.2 yields Corollary 4.3, and the latter provides a threshold on the number of measurements (shots) for estimating each partial derivative during the training. By doing so, the gradient norm at the next point is lower bounded by a 1-epsilon times the gradient norm of the ideal next point using infinite measurements.
>
> **Q3. I suggest providing more examples of when the authors' results are expected to hold (e.g. in chemistry where the Hartree--Fock state is a good approximation of the true ground state, etc.).**
>
> A3: We will add more examples to support the proposed theorems in the final version. For example, the observable of the quantum supervised learning [r5] could be written as the linear combination of Pauli matrix tensor product, which acts on the constant number of qubits that matches Theorem 4.1.
>
> **Reference**
>
> [r1] Anschuetz E R. Critical points in quantum generative models. International Conference on Learning Representations, 2021.
>
> [r2] Liu J, Najafi K, Sharma K, et al. An analytic theory for the dynamics of wide quantum neural networks. arXiv: 2203.16711, 2022.
>
> [r3] You X, Chakrabarti S, Wu X. A Convergence Theory for Over-parameterized Variational Quantum Eigensolvers. arXiv: 2205.12481, 2022.
>
> [r4] Jacot A, Gabriel F, Hongler C. Neural tangent kernel: Convergence and generalization in neural networks. Advances in neural information processing systems, 2018.
>
> [r5] Havlíček V, Córcoles A D, Temme K, et al. Supervised learning with quantum-enhanced feature spaces. Nature, 2019.

---

> > ### Author Response · Authors · 2022-08-09
> > **Addtional response to Q2 and Q3**
> >
> > The Hartree—Fock method is a case of the mean-field theory in quantum chemistry. Thus, we expect that Theorem 4.2 gives nontrivial bounds when the mean-field theory provides a good initial guess, e.g. the quantum many-body problem [r6]. Moreover, we notice that the mean-field theory introduces some interesting results about the training of classical neural networks [r7]. Specifically, Gaussian-initialized classical circuits with the proper variance could be trained quickly [r8]. Although quantum and classical circuits have different parameterizations, we expect that the Gaussian initialization proposed in this work could yield related theories as potential future directions, like the training dynamic of quantum circuits.
> >
> > **Reference**
> >
> > [r6] Amico L, Penna V. Dynamical mean field theory of the Bose-Hubbard model. Physical Review Letters, 1998.
> >
> > [r7] Mei S, Misiakiewicz T, Montanari A. Mean-field theory of two-layers neural networks: dimension-free bounds and kernel limit. Conference on Learning Theory, 2019.
> >
> > [r8] Pennington J, Schoenholz S, Ganguli S. Resurrecting the sigmoid in deep learning through dynamical isometry: theory and practice. Advances in neural information processing systems, 2017.

---

> > > ### Comment · Reviewer_hWp5 · 2022-08-09
> > > **Response to Author Response**
> > >
> > > The authors' response is appreciated. I think the addition of this discussion to the manuscript, along with the clarification of the limitations previously mentioned, would help in making the results and their applicability more clear.

---

> > > > ### Author Response · Authors · 2022-08-09
> > > > **Response to the reviewer**
> > > >
> > > > We thank the reviewer for helpful opinions that improve the quality of our paper! The main text has been refined following the discussion. We refer the readers to lines 199-205 and lines 222-225 for more details.

---

> > > > > ### Comment · Reviewer_hWp5 · 2022-08-09
> > > > > **Response to Author Response**
> > > > >
> > > > > In consideration of the authors' changes, I've updated my score to reflect the improved manuscript.

---

### Official Review · Reviewer_SXHT · 2022-07-07

**Rating:** 7
**Confidence:** 3
**Soundness:** 2 fair
**Presentation:** 3 good
**Contribution:** 3 good

**Summary:**

This paper introduces a new initialization strategy for quantum variational circuits. This gaussian initialization strategy is shown to exponentially increase the upper bound on the gradient, with substantial implications for addressing optimization concerns of medium to large scale quantum machine learning models. Two empirical examples are provided, which show this initialization strategy demonstrates an improvement in performance.

==============================

Note: after author responses score 6 -> 7

**Questions:**

- The heisenberg gamma squared is said to be 1/160 (line 260), however, in the code gamma is set to 1/(8 * n_layer)**0.5. Since n_layer is 10, gamma = 1/sqrt(80) and gamma squared is thus 1/80. There seems to be a factor of two missing, where does this come from?
- Do the optimization improvements from this initialization strategy carry over to gradient free optimizers?
- A variety of gradient based optimizers have been used for QVCs, is there a reason why Adam was chosen over other methods?
- The zero initialization seems to perform very well on LiH (almost exact overlap with gaussian). Is this just a unique aspect of the problem or is zero initialization comparable in performance on large circuits (when the variance becomes very small)?


**Limitations:**

The authors sufficiently addressed the potential negative societal impact of the work.

**Strengths And Weaknesses:**

Pros:
- This work provides a very important improvement in gradient bounds. Given the scale of concerns for optimising QVCs, this is a very exciting result.
- The results for global observables are especially interesting, since previous works have focused on the benefits of local observables to trainability.
- The paper is generally well written and conveys the point effectively
- Circuit diagrams are well done and add to the understandability
- Code is provided in supplementary material, which greatly improves replicability and experimental verification
- The survey of related work is both useful and extensive

Cons:
- The biggest problem is the empirical results. Although these experiments are just examples and the main result is the theoretical proofs, they don’t add as much as they could. Using shot noise, instead of added measurement noise would improve the realism. It would also be beneficial to add examples with more realistic circuit noise (e.g. depolarizing channels). Additionally, showing the gradient norm in Figure 3 (like in Figure 2) would be beneficial
- Citations could be condensed, e.g. “quantum simulations [14, 15, 16, 17, 18, 19, 20, 21, 22, 23]” -> “quantum simulations [14-23]”
- Empirical comparisons to other initialization strategies would be beneficial (e.g. block initialization)
- The horizontal lines on the graphs don’t aid interpretability

---

> ### Author Response · Authors · 2022-08-02
> **Respond to Reviewer SXHT**
>
> We thank the Reviewer for the positive opinions on our paper!
>
> **Q1: The heisenberg gamma squared is said to be 1/160 (line 260), however, in the code gamma is set to $1/\sqrt{8n_{\rm layer}}$. Since n_layer is 10, gamma = $1/\sqrt{80}$ and gamma squared is thus 1/80. There seems to be a factor of two missing, where does this come from?**
>
> A1: We are sorry about the mistake in the code, the gamma squared should be set to 1/160. We have corrected the Fig 2 in the current version.
>
> **Q2: Do the optimization improvements from this initialization strategy carry over to gradient-free optimizers?**
>
> A2: The advantage of our initialization is the large gradient with theoretical guarantees, which is developed aimed at gradient-based optimizers for deep quantum circuits. It is unclear whether the proposed method could surpass other initializations for gradient-free optimizers. However, we would explore the performance of gradient-free methods based on our initialization framework in the future.
>
> **Q3: A variety of gradient-based optimizers have been used for QVCs, is there a reason why Adam was chosen over other methods?**
>
> A3: We have used the gradient descent and the Adam optimizer in the current paper since they have different properties on training, e.g. see Ref.[r1]. Thus, they are good representations among various gradient-based methods to show the effect of the proposed initialization. We are conducting some experiments using other optimizers like the gradient descent with momentum, the Nesterov accelerated gradient, and the Adagrad. The results will be added to the Appendix in the final version.
>
> **Q4: The zero initialization seems to perform very well on LiH (almost exact overlap with gaussian). Is this just a unique aspect of the problem or is zero initialization comparable in performance on large circuits (when the variance becomes very small)?**
>
> A4: The reason for the good performance of the zero initialization on the LiH task is that the zero initialization, which corresponds to the Hartree—Fock state, is a good initial state for small molecules. However, this state has poor capability considering electronic correlations [r2]. Thus, the performance of the zero initialization could be worse on tasks for large molecules or circuits.
>
> **Reference**
>
> [r1] Xie Z, Wang X, Zhang H, et al. Adaptive inertia: Disentangling the effects of adaptive learning rate and momentum. International Conference on Machine Learning, 2022.
>
> [r2] McArdle S, Endo S, Aspuru-Guzik A, et al. Quantum computational chemistry. Reviews of Modern Physics, 2020.

---

> > ### Comment · Reviewer_SXHT · 2022-08-07
> > **Response to Author Response**
> >
> > The authors response to the comments and questions is appreciated. In Q3 it was mentioned that further experiments will be added to the appendix, will these experiments include larger molecules/greater depth as well? The advantages are said to be more apparent (in Q4), which makes them seem like valuable results.

---

> > > ### Author Response · Authors · 2022-08-09
> > > **Respnse to the reviewer**
> > >
> > > We thank the reviewer for the feedback during the discussion stage! We have updated the appendix in the current version with additional experiments on various optimizers and different circuit depths.
> > >
> > > For the Heisenberg model task, we conduct the training using the gradient descent with momentum, the Nesterov accelerated gradient (NAG), and the adaptive gradient (AdaGrad) descent optimizers. We consider different qubit numbers N=15 (Figure 3) and N=18 (Figure 4). The circuit for the latter case contains 20 layers of the $R_Y R_X CZ$ block, which is twice as deep as the original setting. As shown in Figure 3 and Figure 1 in the main text, the performance of GD with momentum and the NAG is similar to that of the Adam optimizer, while the performance of the AdaGrad is similar to the GD optimizer. By comparing Figures 3 and 4, we notice that uniformly initialized circuits converge slower when the qubit number and the circuit depth increase, while Gaussian initialized circuits have similar convergence rates.
> > >
> > > For the quantum chemistry task, we repeat the LiH task in the main text with different layers $L \in (24, 48, 72)$ by stacking the $V_{\rm Givens}$ circuit in Eq.(11). The loss function trained with the gradient descent converges quicker when the depth increases. For the Adam case, circuits with different depths show similar convergence rates. We are sorry that the task of larger molecules with complicated electronic interactions exceeds the capability of our computers. We refer readers to Appendix A.1 and A.2 for more details about these experiments.

---

> > > > ### Comment · Reviewer_SXHT · 2022-08-09
> > > > **Response to Author Response**
> > > >
> > > > Thanks for the information. The changes that will be made from the above comment and the comments to other reviewers will improve the paper and have helped to clarify some of the concerns I had. I have updated the score to reflect the positive outcome of these changes.

---

> > > > > ### Author Response · Authors · 2022-08-09
> > > > > **Response to the reviewer**
> > > > >
> > > > > We thank the reviewer for upgrading the score and providing helpful feedback through the rebuttal stage! Suggestions from reviewers have improved the quality of our manuscript greatly.

---

### Official Review · Reviewer_z13w · 2022-07-15

**Rating:** 6
**Confidence:** 4
**Soundness:** 3 good
**Presentation:** 3 good
**Contribution:** 3 good

**Summary:**

Variational quantum circuits are parametrized models that can be trained to perform mappings using gradient descent methods. In this paper, the authors propose an initialisation strategy to avoid the problem of vanishing gradients that occur when the number of qubits and the circuit depth grow.

**Questions:**

- Theoretical analysis: what happens when the training procedure requires many update steps ? in other words, if the convergence is very slow, is the initialisation technique still useful to avoid vanishing gradients ?
- Experimental analysis: why is the zero initialisation strategy providing competitive results to the Gaussian one ?

**Strengths And Weaknesses:**

- Strengths:
	- Well-written
	- New initialisation: Several initialisation strategies have been studied for classical neural networks but few existing work extend these results to the quantum case. In this work, the authors apply the Gaussian initialisation strategy to variational quantum circuits and study how it may affect the training procedure by providing a theoretical and experimental analysis.
	- Theoretical analysis: The authors start by describing the Gaussian initialisation technique and provide theoretical guarantees in different settings. The first setting corresponds to the case when the circuit architecture is made using trainable 1-qubit gates and the output is projected using local observables. These results are then extended to the global observable case and 2-qubit gates.
	- Experimental analysis: The authors apply their technique to two quantum machine learning problems where they perform numerical simulations to study experimentally the training behaviour of the parameters.

- Weaknesses:
	- The zero initialisation strategy seems to be fine for the performed experiments.

---

> ### Author Response · Authors · 2022-08-02
> **Respond to Reviewer z13w**
>
> We thank the Reviewer for the positive opinions on our paper!
>
> **Q1: Theoretical analysis: what happens when the training procedure requires many update steps? in other words, if the convergence is very slow, is the initialization technique still useful to avoid vanishing gradients?**
>
> A1: Our result about the gradient norm is about the initialization, which is independent of the performance during the training. However, we could provide some helpful intuitive analysis. From the theoretical perspective, the decrease of the loss function in a gradient-based optimization step could be lower bounded by the term that is proportional to the current gradient norm. In practice, we also notice a fast convergence rate if the initial gradient is large. Besides, the reduction of the gradient to zero is a sign of convergence towards stationary points.
>
> **Q2: Experimental analysis: why is the zero initialization strategy providing competitive results to the Gaussian one?**
>
> A2: In the Heisenberg model task, the zero-initialized parameter has similar performance compared to the Gaussian initialized parameter when using noisy gradient descent (Figs. 2b and 2d) and noisy Adam optimizers (Figs. 2f and 2h). Since the gradient at the zero point is zero, the first step of the training using zero-initialized parameters actually generates Gaussian distributed parameters. Thus, it is natural to observe the similar performance of two different initializations, which could be explained by the proposed theory in Theorem 4.1.

---

### Official Review · Reviewer_Wz3c · 2022-07-15

**Rating:** 6
**Confidence:** 5
**Soundness:** 3 good
**Presentation:** 3 good
**Contribution:** 3 good

**Summary:**

The barren plateau phenomenon is a 'vanishing gradient' effect that arises in sufficiently randomly initialized parameterized quantum circuits. Specifically, the norm of the gradient falls exponentially with the number of quantum registers in the circuit. While not a problem for classical neural networks due to efficient gradient estimation procedures, gradients for parameterized quantum circuits are obtained by statistical sampling. Estimating small gradients therefore adds an exponential overhead, eliminating most possible computational advantages.
In recent years there has been an effor to handle this problem by suggesting initializations that are not 'fully random' on the space of circuits. This paper takes the following approach: a (practically reasonable) architecture is chosen and the parameters are initialized from a Gaussian distribution. The main technical message is a proof that if the variance of the normal distributions is chosen as $1/L$ where $L$ is the number of layers, the gradient decays polynomially with the number of qubits $n$ and layers $L$.
Models with the propsoed initialization are evaluated on a variational quantum eigensolver setup to find the ground state of the Heisenberg model and LiH Hamiltonian. On the considered examples, Gaussian initialization appears to outperform the setting where parameters are initialized uniformly.

**Questions:**

- Can there be a general argument about how the results with Gaussian initialization are maintained during the training? Can there be cases where the gradient starts large but reaches a barren region?
- As an associated question, can we reason about the bias induced by keeping the initial unitary close to the identity due to the choice of parameter variance?
- The barren plateau phenomenon was established first for Haar initializations. I did not see a comparison to that setting. Is it similar to the uniformly initialized parameter case.

**Limitations:**

Yes

**Strengths And Weaknesses:**

The main contribution of the paper is the lower bound on the gradient. This proof is very non-trivial and involves techniques and intermediate results, that I think may be of interest when analyzing properties of Gaussian initializations in quantum circuits in general. I have not checked every statement, but I am overall convinced of correctness of the proofs. On a technical level, I think the contribution of the paper is solid. The experimental results also line up with the conclusions drawn, indicating that the phenomenon described may have applicability beyond the theorem setting.

I have two concerns: firstly, due the Gaussian initialization I am not sure that the proof in the paper suffices to say that the gradients are lower bounded *throughout* training since the distribution has morphed from that at initialization. Note that this was not an issue for the original derivation of 'barren plateau' as Haar distributions are invariant under the shifts induced by training. The second is that the restriction to Gaussians with deviation decaying as 1/L, essentially restricts the initialization to a constant neighborhood of the identity. This assumption seems to put more of a bias on the initialization than most existing approaches, and may create the possibility of adversarial problems where convergence is heavily slowed.
Experimentally, the advantage over initializing the parameters to zero seems like an artifact of their being a stationary point at identity, and the addition of some noise to perturb the initial state (in the noisy simulations) seems to remove most observed advantage for the proposed scheme.

---

> ### Author Response · Authors · 2022-08-02
> **Respond to Reviewer Wz3c**
>
> We thank the Reviewer for the general positive opinions on our paper!
>
> **Q1: Can there be a general argument about how the results with Gaussian initialization are maintained during the training?**
>
> A1: We thank the reviewer for the instructive question! The main contribution of this work is an initialization strategy for variational quantum circuits (VQCs), which is guaranteed to be large at the initial stage. We note that the gradient could not maintain large during the training. The norm of the gradient would decay to zero ultimately, which indicates the convergence to stationary points for the non-convex optimization. Thus, we do not need to guarantee the large gradient during the training. The large gradient in the first few iterations could be helpful, since the loss function could decay quickly with gradient-based optimizers. We refer to lines 192-199 in the paper for more discussions.
>
> **Q2: Can there be cases where the gradient starts large but reaches a barren region?**
>
> A2: In general, there could be certain circuits that have the mentioned barren problem. Specifically, barren-plateau-free shallow quantum circuits could have suspicious local minima [r1,r8], which is similar to the mentioned barren region problem for trained parameters. Some papers proposed that the suspicious local minima problem could be solved by using over-parameterized circuits [r2,r3]. Given these results, we think the mentioned barren region problem could be mitigated by using deep circuits towards the over-parameterization stage, where the Gaussian initialization could be helpful.
>
> **Q3: As an associated question, can we reason about the bias induced by keeping the initial unitary close to the identity due to the choice of parameter variance?**
>
> A3: We remark the initialization in Theorem 4.1 requires a fixed Gaussian variance, which is not close to zero arbitrarily but depends on the circuit depth. Besides, the bound in Theorem 4.1 is also independent from the gradient property at the zero point. To verify this result, we have demonstrated the optimization of the Heisenberg task using different Gaussian variances in Appendix A. The training with the proposed parameter variance decays faster than that with a larger or smaller variance.
>
> **Q4: The barren plateau phenomenon was established first for Haar initializations. I did not see a comparison to that setting. Is it similar to the uniformly initialized parameter case.**
>
> A4: The first paper [r4] and several subsequent works [r5,r6] about the barren plateau use Haar initializations, which is convenient for theoretical derivations. However, such initialization is impossible when the circuit structure is given and fixed. Fortunately, the uniformly initialized random circuits are approximately 2-designs [r7], which behave similarly to Haar distributed unitaries. Thus, most previous works [r4-r6] use uniformly initialized parameters in numerical experiments to verify Haar-based theoretical results.
>
> **Reference**
>
> [r1] You X, Wu X. Exponentially many local minima in quantum neural networks. International Conference on Machine Learning, 2021.
>
> [r2] Liu J, Najafi K, Sharma K, et al. An analytic theory for the dynamics of wide quantum neural networks. arXiv: 2203.16711, 2022.
>
> [r3] You X, Chakrabarti S, Wu X. A Convergence Theory for Over-parameterized Variational Quantum Eigensolvers. arXiv: 2205.12481, 2022.
>
> [r4] McClean J R, Boixo S, Smelyanskiy V N, et al. Barren plateaus in quantum neural network training landscapes. Nature communications, 2018.
>
> [r5] Pesah A, Cerezo M, Wang S, et al. Absence of barren plateaus in quantum convolutional neural networks. Physical Review X, 2021.
>
> [r6] Cerezo M, Sone A, Volkoff T, et al. Cost function dependent barren plateaus in shallow parametrized quantum circuits. Nature communications, 2021.
>
> [r7] Harrow A W, Low R A. Random quantum circuits are approximate 2-designs. Communications in Mathematical Physics, 2009.
>
> [r8] Anschuetz E R, Kiani B T. Beyond Barren Plateaus: Quantum Variational Algorithms Are Swamped With Traps. arXiv: 2205.05786, 2022.

---

### Author Response · Authors · 2022-08-02
**General Response**

We sincerely thank all reviewers for their helpful questions and constructive suggestions! It is glad to see that all reviewers give positive feedback on our work. We will refine our paper and address all concerns and suggestions in the final version.

---

### Author Response · Authors · 2022-08-10
**Summary of improvements**

We would like to express our appreciation to all reviewers for their constructive questions and helpful feedback through the rebuttal period! The manuscript and the appendix have been refined following suggestions. Here we summarize the main improvements.

1. We have added more discussions about conditions when proposed theorems lead to meaningful results. We refer readers to lines 199-205 and lines 222-225 for more details.

2. We have conducted additional experiments considering various optimizers and different circuit depths. These results have been illustrated in Appendix A of the current version.

---

### Meta-Review · Area_Chair_qmnu · 2022-08-24

**Recommendation:** Accept
**Confidence:** Certain

**Metareview:**

The authors propose a new random initialization of quantum neural networks which could avoid generating vanishing gradients.  Specifically, the new random (Gaussian) initialization scheme will depend on the shape of the ansatz so that the norm of the gradient decays at most polynomially when the qubit number and the circuit depth increase.  This finding is also supported by the associated empirical study.  The reviewers consider this an important step toward the understanding of the trainability of variational quantum circuits. However, some limitations of the proposal are also discussed in the reviews, and we hope the authors can make an explicit discussion of these limitations in the final version.

**Award:**

No

---

### Decision · Program_Chairs · 2022-09-14

Accept